# Immunotherapy and Radiotherapy as an Antitumoral Long-Range Weapon—A Partnership with Unsolved Challenges: Dose, Fractionation, Volumes, Therapeutic Sequence

**Camil Ciprian Mireștean** [1,2] , **Roxana Irina Iancu** [3,4,*] **and Dragoș Teodor Iancu** [5,6]

1    Department of Medical Oncology and Radiotherapy, University of Medicine and Pharmacy of Craiova, 200349 Craiova, Romania
2    Department of Surgery, Railways Clinical Hospital, 700506 Iasi, Romania
3    Oral Pathology Department, "Grigore T. Popa" University of Medicine and Pharmacy, 700115 Iasi, Romania
4    Department of Clinical Laboratory, "St. Spiridon" Emergency Hospital, 700111 Iaşi, Romania
5    Department of Medical Oncology and Radiotherapy, "Grigore T. Popa" University of Medicine and Pharmacy, 700115 Iasi, Romania
6    Department of Radiation Oncology, Regional Institute of Oncology, 700483 Iasi, Romania
*    Correspondence: roxana.iancu@umfiasi.ro; Tel.: +40-232-301-603

**Abstract:** Immunotherapy, the modern oncological treatment with immune checkpoint inhibitors (ICIs), has been part of the clinical practice for malignant melanoma for more than a decade. Anti-cytotoxic T-lymphocyte antigen 4 (CTLA4), anti-programmed cell death Protein 1 (PD-1), or anti programmed death-ligand 1 (PD-L1) agents are currently part of the therapeutic arsenal of metastatic or relapsed disease in numerous cancers; more recently, they have also been evaluated and validated as consolidation therapy in the advanced local stage. The combination with radiotherapy, a treatment historically considered loco-regional, changes the paradigm, offering—via synergistic effects—the potential to increase immune-mediated tumor destruction. However, the fragile balance between the tumoricidal effects through immune mechanisms and the immunosuppression induced by radiotherapy means that, in the absence of ICI, the immune-mediated potentiation effect of radiotherapy at a distance from the site of administration is rare. Through analysis of the preclinical and clinical data, especially the evidence from the PACIFIC clinical trial, we can consider that hypofractionated irradiation and reduction of the irradiated volume, in order to protect the immune-infiltrated tumor microenvironment, performed concurrently with the immunotherapy or a maximum of 2 weeks before the start of ICI treatment, could bring maximum benefits. In addition, avoiding radiation-induced lymphopenia (RILD) by protecting some anatomical lymphoid structures or large blood vessels, as well as the use of irradiation of partial tumor volumes, even in plurimetastatic disease, for the conversion of a "cold" immunological tumor into a "hot" immunological tumor are modern concepts of radiotherapy in the era of immunotherapy. Low-dose radiotherapy could also be proposed in plurimetastatic cases, the effect being different (modeling of the TME) from that of high doses per fraction irradiation (cell death with release of antigens that facilitates immune-mediated cell death).

**Keywords:** radiotherapy; immunotherapy; abscopal; synergy; dose; fractionation; sequence; SRT; SBRT; hypo-fractionation

---

## 1. Introduction

The approval in 2011 of anti-cytotoxic T-lymphocyte antigen 4 (CTLA4) therapy with ipilimumab for advanced melanoma opened the era of clinical implementation of a new class of agents with an antineoplastic role [1]. The therapeutic results of immune checkpoint inhibitors (ICI) have essentially changed the evolution and prognosis of cancer; there are situations and cases in which immunotherapy has proven superior to traditional

oncological therapies. Preclinical data suggest a possible benefit of combining radiotherapy with immunotherapy in order to obtain synergistic results [2].

The treatment of ionizing radiation delivered to the tumor site is effective on both the tumor and the tumor stroma. The cell lesions produced by irradiation expose tumor antigens to the effect of the host's immune system, making them targets of immune surveillance, but also of cytotoxic T lymphocytes [1]. The effect of irradiation on the antitumor immune response is amplified and the modulation induced on the tumor microenvironment can also facilitate the recruitment and infiltration of immune cells. Blocking ICIs can contribute to an amplified effect of the antitumor immune system, thus demonstrating the premises of a successful partnership between radiotherapy and immunotherapy [1,3,4].

Non-small cell lung carcinoma (NSCLC) is the tumor that validated the preclinical data supporting the benefit of the synergistic association between radiotherapy and immunotherapy. The combination of immunotherapy and radiotherapy has become a therapeutic standard, the consolidation treatment in NSCLC in the case of stable locally advanced disease after radiochemotherapy and immunotherapy with durvalumab. Even if the association of immunotherapy with radiotherapy is a future strategy with immense potential, there are still questions that fundamental and clinical research will have to answer: therapeutic sequence (concurrent or sequential treatment; induction or consolidation), radiation dose, fractionation (standard fractionation, hypo-fractionation, stereotactic body radiotherapy (SBRT)), volumes (including or not the tumor microenvironment or elective nodes treatment), adding or not chemotherapy [5].

In this review, we propose to briefly present aspects that can help us elucidate the still unresolved questions related to radiation doses, fractionation, target volumes, and therapeutic sequence for the large-scale implementation of the synergistic association of radiotherapy and immunotherapy.

## 2. The Effect of Radiotherapy on the Antitumor Immune Response

Historically, radiotherapy was considered to induce cell death by producing double-stranded DNA damage, the mechanisms involved being multiple (apoptosis, mitotic catastrophe, and senescence). Following the destruction of malignant cells by irradiation, a series of products result that trigger reactions of the immune system with both stimulatory and suppressive effects. The presentation of the tumor antigen mediated by irradiation has the effect of activating dendritic cells (DCs), increasing the degree of lymphocyte infiltration of the tumor, but also control cell signaling [1,5,6]. Irradiation also has the effect of cross-presentation of the antigen to T cells via DCs and the secretion of inflammatory cytokines. Irradiation has the ability to modulate the expression of interferon 1 (IFN-I), resulting in the activation of CD8+ lymphocytes. Chemokines (C-X-C motif) ligand 10 (CXCL10) and CXCL16 and vascular cell adhesion molecule 1 (VCAM-1) and intercellular adhesion molecule (ICAM), vascular adhesion molecules, are also modulated by irradiation and participate in the process of tumor detachment from the substrate, invasion, and metastasis. By upregulating the FAS pathway by irradiation and by the induction of sub-lethal DNA lesions the tumoricidal action of CD8 improves, mediating apoptosis by caspases (3, 6, and 7). Thus, the signaling mechanisms are also involved in the tumor response to irradiation. The upregulation of FAS ligand (FAS-L) on endothelial cells could have the final effect of T-lymphocyte death and could promote tumor growth and immunosuppression, blocking cytotoxic T-lymphocyte activation and maturation. The induction of an immunosuppressive microenvironment by irradiation is also potentiated by the recruitment of regulatory T cells (Tregs), tumor associated macrophages (TAMs), and myeloid derived suppressor cells (MDSCs). The recruitment of MDSCs is mediated by neo-angiogenesis via Hypoxia-Inducible Factor-1 (HIF-1). Thus, irradiation should be seen as a treatment with a double effect: potentiation of the immune antitumoral effect and immunosuppression. Identifying the best strategies that will exploit the antitumor effect and limit tumor growth is the goal of present and future research [7,8]. The extracellular matrix (ECM) is more rigid in tumor cells; stiffness and density are associated with unfavorable prognosis, tumor aggressiveness,

unfavorable response to treatment, and especially to irradiation. Radiotherapy also has the property of altering the mechanical properties of the ECM [7,8].

### 3. Dose and Dose Fraction Factors

An irradiation dose of >8Gy will have the effect of reducing blood flow in the tumor and will precipitate apoptosis of the vascular endothelium. In addition, high doses per fraction will also have an effect on T lymphocytes, producing apoptosis. Doses per fraction close to the standard value of 2Gy will have the effect of recruiting and increasing the intensity of T-lymphocyte trafficking, but will lead to the inactivation of natural killer (NK) lymphocytes and infiltrate the tumor with factors with an immunosuppressive effect, such as regulatory T-lymphocytes and MDSCs. Thus, radiotherapy can contribute to the generation of an immunosuppressive peritumoral microenvironment [5,9].

The immunological response to localized irradiation with a mediating role of systemic anticancer immune activation is called the abscopal effect and is strictly dependent on the immunogenic effects of radiotherapy [2,10]. Reynders and collaborators identified, from an analysis of the literature focused on the abscopal effect from 1960 to 2014, 23 case reports, 1 retrospective study, and 13 preclinical trials, some also using immune modulation therapies. The median radiation dose used was 32Gy and the dose per fraction varied from 1.2Gy to 26Gy. The time until the abscopal effect was established varied from less than 1 month to 24 months. The authors note the synergistic effect demonstrated by preclinical studies, an effect that seems to have been minimized as a clinical importance, although data on abscopal potentiation by adding a systemic immunomodulatory agent to irradiation are quite old [10,11]. A systematic review of the literature identified a total of 46 cases reported from 1969 to 2014. In that case, the median dose reported was 31Gy, and the average time until the abscopal effect was reported was 2 months. Even if there are reports of the abscopal phenomenon, the dual effect of irradiation (immune-potentiator and immune-suppressor) makes its manifestation exceptional in the absence of a systemic treatment with the potential to inhibit antitumor immune mechanisms. The association of a single ICI with irradiation was associated with the induction of the abscopal effect in 24% and 37% when anti-CTLA4 and anti-programmed death-ligand (PD-L1) therapies were used, respectively. In the case of a double combination of anti-CTLA4 and anti-PD-L1, the local response rates were 71%, and the abscopal response rates were 41%. These data justify the growing interest in clinical trials to answer the uncertainties related to the optimal sequence of the immunotherapy–radiotherapy combination [12].

The report of Ozpiskin and colleagues in 2018 identified 67 studies associated with anti-CTLA4 therapy, 182 studies associated with anti-programmed cell death Protein 1 (PD-1), and 186 studies associated with anti-programmed death ligand 1 (PD-L1) agents with radiotherapy according to data obtained from clinicaltrials.gov. The authors also mention the need to elucidate the mechanisms of the antitumor response produced by irradiation in combination with ICI, but identify CXCL16, CTLA-4, CD47, humanized epidermal growth factor receptor (HER), PD-1, and PD-L1 as key players in the "immune escape" of tumor cells; the results showed an unfavorable response of malignant tumors to irradiation [13]. Four years later, Khalifa et al. reported the existence of 700 clinical studies that evaluate the association of radiotherapy with immunotherapy, but also draw attention to the paradigm shift by transforming a historically considered loco-regional treatment into a treatment with the effect of activating a systemic immune response [14].

The non-randomized single-arm phase II study NCT04951115 including subjects with previously untreated stage IV small cell lung cancer proposes five days of non-ablative SBRT followed by four cycles of chemo-immunotherapy (etoposide + carboplatin or cisplatin + durvalumab). Durvalumab can also be administered after four cycles until disease progression. The radiation dose of 6 Gy $\times$ 5 fractions is administered from the first day of chemo-immunotherapy at several tumor sites. The trial conceptually exploits the benefit brought by a sub-ablative dose of radiation added to chemo-immunotherapy for treatment-naïve, extended-stage small cell lung cancer patients [15,16].

Head and neck cancers, considered poorly responsive to single-agent immunotherapy in the metastatic and recurrent disease stages, were analyzed in a randomized phase II trial of nivolumab versus nivolumab plus SBRT that aimed to demonstrate synergistic radiotherapy–immunotherapy in HNSCC. Cases with at least two metastatic lesions were included; at least one of them had to be measurable according to RECIST version 1.1. The group was stratified by human papilloma virus (HPV) status and randomized to receive nivolumab 3 mg/kg intravenously every 2 weeks and nivolumab (same regimen) plus SBRT, three fractions, 9Gy per fraction on a single lesion. The abscopal effect was evaluated by RECIST evaluation of the non-irradiated lesion; the objective was ORR. Lower levels of PD-L1 and CD8 infiltrating T-cells were identified in adenoid cystic carcinoma. The use of an immunotherapy-radiotherapy combination did not produce systemic effects in a phase II study, but in patients with progressive disease before treatment, a high rate of stationary disease was obtained [17]. The regimen that associates pembrolizumab with radiotherapy (30Gy in five fractions) is feasible and well tolerated, but the study did not reach its survival objectives. MYB/NFIB translocation and PD-L1 expression were associated with the local response rate to the combined treatment [18,19].

The hypothesis that low-dose irradiation can reprogram the tumor microenvironment and that it has the ability to amplify the effect of immunotherapy was exploited in the phase II study that evaluated the benefit of combining the double inhibition of PD-L1 with durvalumab and CTLA-4 with tremelimumab as a single treatment or combined with radiotherapy in patients with NSCLC refractory to single-agent immunotherapy. The study enrolled metastatic NSCLC cases with an ECOG performance index of 0 or 1 who progressed on PD(L)-1 immunotherapy. The second line of immunotherapy included a maximum of 13 cycles of durvalumab and a maximum of four cycles of tremelimumab at a dose of 1500 mg, respectively, 75 mg every 4 weeks [20]. An irradiation dose of 0.5Gy repeated two times a day during the first 2 days of the first five cycles of immunotherapy or a hypofractionated regimen with a total dose of 24Gy in three fractions of 8Gy only during the first cycle have been proposed as an association to double immune blockade. In the case of hyperfractionated irradiation, an interval of one week existed from the administration of the durvalumab–tremelimumab doublet to the first dose of irradiation. The study was stopped prematurely, as the preliminary results showed no benefit. In the group that combined irradiation with low doses, there was also a case of grade 5 toxicity (death being caused by respiratory failure). In addition, the toxicity rate was significantly higher in the group that combined irradiation with ICI therapy (19% for low-dose irradiation vs 15% for hypofractionated radiotherapy vs 4% for double ICI blockade alone). A critical analysis of the study proposed by Schoenfeld et al. noted the value of this concept of using low-dose radiotherapy to modulate TME, but mentions the need to irradiate all metastatic lesions if this concept is used [21]. Ochoa-de-Olza suggests that irradiation with high doses has a tumoricidal effect, but also releases danger molecules that lead to the recruitment of immune cells and induce a systemic response to tumor antigens with a systemic effect and a protective effect against recurrence and metastasis. However, the small doses have the effect of reprogramming the TME; both irradiation regimes have the potential to refine future clinical practice, with conceptual association between radiotherapy and ICI being a perspective for the future [22].

## 4. The Volume Factor

The new concept mentioned by Ozpiskin and colleagues includes the possibility of irradiating partial volumes of the tumor or metastases for the strict purpose of activating the immune system and transforming the tumor from an immunologically "warm" tumor into a "cold" tumor [13]. The concept of immunological-clinical target volume (ICTV) and irradiation of partial volumes, contradicting the classical theory of target volumes, was previously postulated by Mireștean and collaborators in the article 'Synergies Radiotherapy-Immunotherapy in Head and Neck Cancers. A New Concept for Radiotherapy Target Volumes-"Immunological Dose Painting" [23]. Adjustments in the classical theory of target

volumes also include the indication of elective irradiation of lymph nodes. The migration of dendritic cells (DC) in lymphatic ganglions for priming of the CD8+ Lymphocyte T could be affected by elective irradiation, and the sterilization effect of possible micrometastases is associated with an immunosuppressive effect with the potential to reduce the tumoricidal effect of irradiation [5,13].

Reducing the setup margins of radiotherapy by improving the imaging guidance technique, as well as an even a more extreme theory that assumes the omission of irradiation of the microscopic disease, and that the benefit of saving lymphocytes with an antitumor immune effect is greater than the benefit of sterilizing the microscopic disease are hypotheses to be taken into account in the future immunotherapy–radiotherapy therapeutic association [5]. The preclinical results still advocate for the irradiation of the tumor microenvironment, mentioning the survival of the effector immune cells, but the irradiated cells still displayed antitumor action. Irradiation can reprogram T cells, an effect mediated by Transforming Growth Factor β (TGF-β), so that they express signatures related to the epithelial–mesenchymal transition, adhesion, and angiogenesis. Radiation-induced lymphopenia (RILP) is a factor with the potential to reduce antitumor immunity by reducing the number of circulating CD4+ and CD8+ T lymphocytes. A dose of 3Gy has the effect of reducing the number of circulating lymphocytes by 90%, but the effect decreases significantly, doses of 2Gy and 0.5Gy are associated with a reduction of 50% and 10%, respectively. There is an obvious need to protect and include as risk organs both the lymphoid tissues and the large vessels that carry large blood volumes or have high blood velocity flows. Techniques such as Intensity Modulated Radiation Therapy should include additional dose constraints and new structures defined for cases that could benefit from immunotherapy [5,14,24].

## 5. Immunotherapy–Radiotherapy: Treatment Sequence

Data from the phase III PACIFIC trial, a study in which immunotherapy with durvalumab was administered as maintenance after chemoradiotherapy for patients with unresectable stage III lung cancer, showed impressive results in survival. The initiation of durvalumab within the first 14 days after completion of radiotherapy was associated with a greater survival benefit than the initiation of immunotherapy between 14 and 42 days after irradiation [25]. A subsequent exploratory analysis of the PACIFIC trial data demonstrated a durable response with OS and PFS rates of 49.6% and 35.3% at 4 days of treatment for patients randomized to the durvalumab booster chemoradiotherapy and immunotherapy arm [25,26]. Data from the KEYNOTE-01 trial demonstrate a long-term interaction between radiotherapy and immunotherapy after an initial interaction on days 2–7. An updated analysis of the KEYNOTE-001 clinical trial data proposed by Shaverdian et al. mentions clearly superior treatment results for patients who previously received radiotherapy. A superior PFS and OS (6.3 months and 11.6 months) was obtained for the cases that had a history of irradiation, compared to a PFS of 2 months and an OS of 5.3 months for the cases without previous radiotherapy. A massive abscopal effect with release of non-antigens and tumor exposure to the immune response was the possible explanation proposed by Liu and collaborators [27–29].

Data regarding stereotactic radiosurgery (SRT) in a preclinical model did not show benefit in the case of using a single dose of 10Gy before the start of immunotherapy. Vascular permeability has been used as a surrogate marker for tumor immune activation in preclinical models. An increase in this biomarker was recorded after the first 24 hours after irradiation, and after 3-10 days a reduction to the baseline level was observed [19]. A study that combined immunotherapy with CTLA-4 inhibitors in malignant melanoma with brain metastases showed better results in OS if single-fraction stereotactic radiosurgery (SRS) was performed before ipilimumab administration or concomitantly, relative to the situation in which immunotherapy was initiated before irradiation. The study also highlights a slight tendency towards a benefit of the concurrent administration of radiosurgery and immunotherapy [30–32].

The concept of synergy of chemotherapy and immunotherapy plus concurrent radiotherapy is exploited in a study (NIRVANA-Lung) (ClinicalTrials.gov identifier, NCT03774732). The trial is based on the results of two randomized Phase II studies that demonstrate the concept that concurrent irradiation with pembrolizumab significantly improves the therapeutic benefit in terms of survival compared to immunotherapy alone. The study includes both squamous and non-squamous advanced NSCLC cancers. The NIRVANA-Lung randomized trial is considered the first phase III trial of concurrent RT and pembrolizumab associated with chemotherapy [33]. RTOG 3505, another randomized phase III study, evaluated the combination of concurrent chemoradiation followed by immunotherapy in cases of locally advanced NSCLC. The proposed chemotherapy protocol includes cisplatin and etoposide and radiotherapy in a total dose of 60Gy, and the group that will receive active treatment with nivolumab 240 mg I.V. every 2 weeks for up to 1 year. The aim of the study is to evaluate OS and PFS, but also the quality of life and the toxicity profile in relation to the PD-L1 status, considering the 1% value as the cutoff. The study aims to randomize a total number of 550 patients [34].

The NICOLAS study coordinated by Peters et al. was the first completed phase II study to evaluate the safety and then the efficacy of the addition of pembrolizumab to platinum-based chemotherapy and concurrent radiotherapy for a total dose of 66 Gy in 33 fractions. The study included 74 patients with stage III NSCLC. The median OS rate at 2 years was a median OS of 63.7%, with stage IIIA associated with an OS of 81% and stage IIIB with an OS of 56%. A PFS value of at least 45% in one demonstrates the efficacy of the regimen, even if the final OS and PFS data are higher for other studies involving the same stages of the disease [35].

For advanced NSCLC cases (stage II–III unresectable or inoperable) with PD-L1 expression > 50%, the NRG-LU004 trial proposed replacing chemotherapy with immunotherapy (1500 mg durvalumab) concurrently with radiotherapy. Immunotherapy was administered once a month for 1 year and accelerated fractionated radiotherapy (ACRT) at 60 Gy in 15 fractions and 60 Gy in 30 fractions followed by a safety hold of 90 and 60 days, respectively. All 13 cycles of immunotherapy were completed by 24% of patients and grade III toxicities were identified in both groups. The deaths in each arm were not related to therapy. A regimen that replaces chemotherapy with concurrent immunotherapy with durvalumab for cases with high PD-L1 values is feasible, as it is necessary to validate the regimen in phase II and III trials [36].

## 6. Conclusions

The association of radiotherapy, a treatment considered loco-regional, with immunotherapy results in a changed paradigm, offering a potential to increase the effect of the antitumor immune system through synergistic effects. However, the fragile balance between immune-mediated antitumor effects and radiotherapy-induced immunosuppression means that, in the absence of immune checkpoint inhibitors (ICIs), the immune-mediated potentiation effect of distant (abscopal) radiotherapy is rare. Through analyzing the preclinical and clinical data, especially the evidence from the PACIFIC clinical trial, we can consider that maximum benefits may be achieved by hypofractionated irradiation, limiting the irradiated volume to protect the immune infiltrated tumor microenvironment, performed concurrently with the treatment based on an ICI or a maximum of 2 weeks before the start of the immunotherapy. Additionally, avoiding radiation-induced lymphopenia (RILD) by protecting some lymphoid anatomical structures or large blood vessels, as well as the use of irradiation of partial tumor volumes even in plurimetastatic disease for the conversion of a "cold" immunological tumor into a "hot" immunological tumor are modern concepts of radiotherapy in the era of immunotherapy. Low-dose radiotherapy could also be proposed in plurimetastatic cases, as the effect is different (modeling of the TME) from that of the high doses per fraction used in radiotherapy (cell death with release of antigens that facilitates immune-mediated cell death). It is definitely an era in which immunotherapy and radiotherapy can establish a long-term partnership, but certain aspects related in particular

to the choice of target volumes and the potential negative effect of RILD definitely deserve the same attention as factors such as dose/volume/fractionation and therapeutic sequence.

**Author Contributions:** Conceptualization, C.C.M., R.I.I. and D.T.I.; methodology, C.C.M.; validation, R.I.I. and D.T.I.; writing—original draft preparation, C.C.M.; writing—review and editing, R.I.I. and D.T.I.; supervision, R.I.I. and D.T.I. All authors have read and agreed to the published version of the manuscript.

**Funding:** This research received no external funding.

**Conflicts of Interest:** The authors declare no conflict of interest.

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
