# Peer review of "Immunotherapy and Radiotherapy as an Antitumoral Long-Range Weapon—A Partnership with Unsolved Challenges: Dose, Fractionation, Volumes, Therapeutic Sequence"

_curroncol, doi:10.3390/curroncol29100580_

Round 1

Reviewer 1 Report

The subject of the review is interesting, but the manuscript could not provide a clear-cut case.

1.       The title is confusing; what are the dilemmas? Dose, fractionations, and others are challenges that need more study. The title needs to be changed.   

2.       Many statements and paragraphs without references, examples, lines 37-49, 137-157, 193-203 and many others.

3.       In lines 186-190, the progression-free survival is longer than the overall survival!!!!

4.       No graph, table, diagram, or mechanist shown in the manuscript can explain any review point.

5.       The review does not add any more meaningful information than the 2018 review shown in reference 1 in your manuscript.

6.       The English of the whole manuscript needs revision. 

Author Response

Dear Revier,

Thank you for the effort and time allocated to evaluate the manuscript and for the relevant comments.

We have carried out a general revision of the article, correcting certain gaps in reference citation omitted by mistake (including even one of our team article). We also completed the manuscript with new data, increasing the number of bibliographic references and corrected some inappropriate expressions. I also corrected the mistake of mentioning a longer PFS than OS. We have added recent trials and data related to low-dose radiotherapy. Not being a lengthy review but one that touches on several issues that we consider important not to be omitted (including RILD and the choice of target volumes that are less evaluated in studies than dose fractionation and sequence) we have not included tables.

In the hope that you will appreciate the new version of the manuscript, we are waiting for new suggestions to improve it.

Best regards,

Camil Mirestean

Reviewer 2 Report

The review manuscript titled “Immunotherapy & Radiotherapy as antitumoral long-range weapon – A partnership with dilemmas: dose, fractionation, volumes, therapeutic sequence” submitted by the authors did not make a comprehensive review and summary of the available literature. The current manuscript with limited information from only 17 references is far from robust the content. More reviews must be performed.

Author Response

(The authors gave the same response as above.)

Round 2

Reviewer 2 Report

I respect your work. But I keep my concerns as before, sorry for I could not recommend a review manuscript like this.  Good luck.

Author Response

Dear Reviewer,

We appreciate your honesty and opinion about our article.
We also thank you for taking the time to review the manuscript

Best regards,
Camil Mirestean